# Fabrication of Porous SiC by Direct Selective Laser Sintering Effect of Boron Carbide

**Rongzhen Liu** [1,2]**, Gong Chen** [1]**, Yudi Qiu** [1]**, Peng Chen** [1]**, Yusheng Shi** [1,]*****, Chunze Yan** [1] **and Hongbin Tan** [2,]*****

[1] State Key Laboratory of Materials Processing and Die & Mould Technology, School of Materials Science and Engineering, Huazhong University of Science and Technology, Wuhan 430074, China; rongzhenliu@hust.edu.cn (R.L.); chengong@hust.edu.cn (G.C.); qyd5221234@126.com (Y.Q.); thisiscp@163.com (P.C.); c_yan@hust.edu.cn (C.Y.)

[2] State Key Laboratory of Environment-Friendly Energy Materials, School of Materials Science and Engineering, Southwest University of Science and Technology, Mianyang 621010, China

***** Correspondence: shisyusheng@hust.edu.cn (Y.S.); hb-t@163.com (H.T.); Tel.: +86-027-8755-8188 (Y.S.); +86-816-241-9022 (H.T.)

**Abstract:** Additive manufactured porous SiC is a promising material applied in extreme conditions characterised by high temperatures, chemical corrosion, and irradiation etc. However, residual Si's existence deteriorates its performance and limits its application in harsh environments. In this study, $B_4C$ was introduced into the selective laser sintering process of SiC, and its effects on forming ability, pore parameters, microstructure, and phases were investigated. The results showed that when $B_4C$ was added, the processing window was enlarged. The minimum energy density was reduced from 457 J/cm$^2$ to 214 J/cm$^2$ when the content of $B_4C$ reached 15 wt%. Microstructure orientation was enhanced, and the residual silicon content was decreased from 38 at.% to about 8 at.%. Small pores were turned into large pores with the increase of $B_4C$ addition. The findings indicate that the addition of $B_4C$ increases the amount of liquid phase during the laser sintering process of silicon carbide, improving the SiC struts' density and reducing the residual silicon by reacting with it. Therefore, the addition of $B_4C$ will help improve the application performance of selected laser-sintered silicon carbide under extreme conditions.

**Keywords:** selective laser sintering; porous SiC; phase evolution; $B_4C$; microstructure

## 1. Introduction

Porous SiC shows many excellent characteristics such as radiation resistance, good chemical stability, and good high-temperature mechanical properties. It has been widely used in extreme conditions such as the aerospace, nuclear, and chemical industries [1]. Many methods have been developed to fabricate porous SiC, including the template, pore former, in situ reaction, foaming agent, freeze-drying method, etc. [2–7]. Although processes such as extrusion molding and gel casting can produce some porous SiC structures, since chemical bonds of SiC are mainly covalent, the manufacture of complex-shaped porous SiC parts is still a big challenge [8,9].

With the development of technology, porous components used in extreme environments characterized by high temperature, corrosion, and radiation have multiple property requirements such as structural strength, service life, heat exchange performance [10]. For example, high filtration flux requires the filter component to have high porosity and large pore size. In contrast, high strength requires high density and smaller pore size. Structural design is one of the most promising methods for solving the contradictions between different performances. Traditional fabrication methods rely on molds, and their structural design is limited by molds and lacks flexibility. Therefore for porous SiC fabricated by conventional methods, it has been challenging to meet the application materials' requirements in extreme conditions. The additive manufacturing of ceramics provides new techniques

and ideas for solving the above problems. Stereolithography (SL) technology and selective laser sintering (SLS) are the most common methods for the additive manufacturing of SiC parts. SiC fabricated by extrusion-based additive manufacturing and 3DP technology were also reported [11,12].

The stereolithography method is the earliest developed technology and is also one of the most widely commercialized technologies due to its high accuracy and ease of forming. However, due to the high absorption rate of SiC to ultraviolet light, the stereolithography process of SiC is complicated. Therefore, Tian et al. first developed an indirect strategy for stereolithography of SiC. This method converts the resin into carbon through a debinding process and then turns the carbon body into SiC parts through a reaction sintering process [13]. Because the infiltration process of liquid-phase silicon would fill all the pores inside the green body, it became challenging to prepare a SiC component with a microporous structure by this method. Ding et al. used SiC and light-curing resin as the raw material to prepare SiC complex parts by stereolithography, which proved the feasibility of forming SiC by light-curing technology [14]. However, the forming efficiency is low, and the size of the parts is limited. SiCw/SiCp-reinforced SiC composites structures with good properties were also prepared by the method [15]. Zhang et al. obtained a SiC lattice structure by digital light processing followed by a silicon infiltration process [16]. The lattices strength can reach 262.6 MPa and with curved geometry. However, due to the limitation of forming efficiency and the debinding process, the SiC parts' size is limited.

The SLS process is one of the most promising methods for fabricating large-scale SiC ceramic parts. At present, there is a lot of research on the SLS process of SiC using polymers as sintering aids. For example, Liu Kai et al. fabricated complex SiC parts with the SLS method using SiC powder and phenol-formaldehyde resin [17]. Zhou Peng etc., investigated the influence of PVB on the SLS process and pointed out that the amount of PVB on the SiC surface has a significant impact on forming [18]. Since the SLS process uses polymers to bond ceramic particles, both polymer removal and sintering of SiC are needed in the subsequent procedures, leading to the problem of long forming cycles. The SiC samples prepared by the SLS process usually require the Si infiltration process, which often brings other issues. For example, Stevinson et al. studied silicon's overfilling phenomenon after the silicon infiltration in the SiC components prepared by SLS. They pointed out that this phenomenon is related to the volume expansion of silicon during the solidification process [19].

As we know, excessive residual silicon content will reduce the materials high-temperature mechanical properties and reduce the radiation resistance of SiC materials under extreme conditions. Therefore, to lower the residual silicon content, the carbon impregnation process was applied. Jin et al. used the polymer infiltration pyrolysis (PIP) process after SLS and fabricated SiC parts with nearly no residual silicon [20]. Meyers et al. also adopted similar methods and obtained SiC parts with a residual Si content of about 12% [21]. The infiltration pyrolysis process needs to be repeated many times, which significantly reduces the formation efficiency.

In addition to carbon addition, much of the current literature on reaction sintering of SiC has shown that boron carbide can react with Si and reduce the residual Si content in SiC [22,23]. Compared with the addition of polymer additives, the addition of $B_4C$ could benefit the direct SLS process of the SiC ceramic part. As complex structures can be formed directly, boron carbide will enable SLS of SiC to have higher forming efficiency and is expected to reduce the amount of residual silicon. Therefore, in this study, boron carbide addition in the SLS process of SiC is investigated, especially the influence on microstructure evolution, pore parameters, and residual silicon.

## 2. Materials and Methods

### 2.1. Experimental Raw Materials

In this experiment, SiC powders with 98% purity were used, with a mean particle size of about 20 μm, and $B_4C$ powder with a purity of 99.7% (D50 = 20 μm) was used. The SiC

powder is produced by Shandong Weifang Huarong Ceramic Materials Co., Ltd. (Weifang, China), and its phase is $\alpha$-SiC. The $B_4C$ powder used is produced by Mudanjiang Diamond Boron Carbide Co., Ltd. (Heilongjiang, China). Their morphologies are shown in Figure 1.

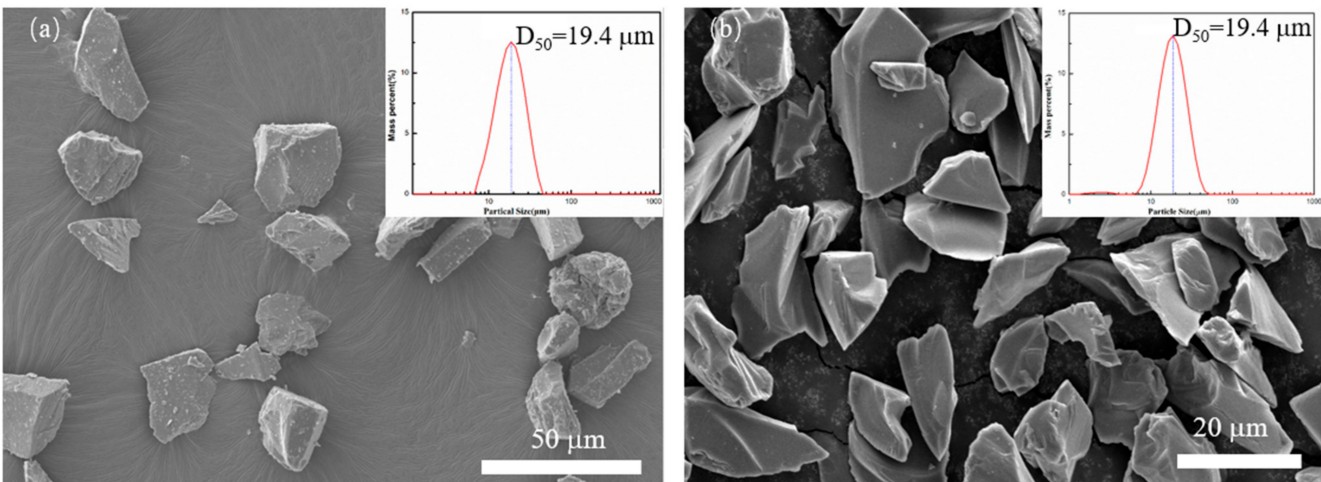

**Figure 1.** Particle size distribution and scanning electron microscopy (SEM) photo of applied powder (**a**) SiC, (**b**) $B_4C$.

### 2.2. Powder Treatment and Forming Process

Ethanol was used as a dispersant. SiC and $B_4C$ powders were mixed by high-energy ball milling. The containers for balling were made of polytetrafluoroethylene (PTFE), and the balls were made by $ZrO_2$. The mass ratio of the ball to the powder was 1:1, the ball milling speed was 200 rpm, and the time was 14 h. After the powder was dried, the mixed powder was adopted for direct selective laser sintering (dSLS) forming. The equipment used for the SLS process was an M250 (independently developed by Huake 3D, Wuhan, China). The default layer thickness was set to 0.1 mm, and the hatch space was 0.05 mm. During the forming process, the forming cavity was filled with argon gas, and the pressure of the argon gas was 1 atm to prevent the specimen from being oxidized during the forming process. During laser sintering, the mixed powder was put into a bunker and then paved by the roller to form the powder layer. A 316 L steel plate was adopted as a base plate. The powder was then laser sintered to form the porous sample. Square samples with a dimension of 8 mm $\times$ 8 mm with a thickness of about 5 mm) were fabricated to evaluate the integrity and line shrinkage during the laser sintering process. After laser sintering, the samples were carefully removed from the base plate by wire cutting. This type of sample was also used for morphology and phase analysis.

### 2.3. Characterization

Samples were cut into small pieces about 2 mm for the mercury intrusion test. The mercury intrusion method measured the material's pore size distribution (AutoPore 9500, Mike Instruments, Bellevue, WA, USA). Mastersizer 3000 (Malvem Co., Ltd., Worcestershire, UK) was applied to determine the powder's particle size distribution. The morphology of the powder particles and the formed samples were determined with the field emission scanning electron microscope (JSM-7600 F of JEOL Co., Ltd., Akishima, Tokyo, Japan). The phase of raw materials and samples was determined by X-ray diffractometer (XRD-7000S equipment of Shimadzu Corporation of Japan, Tokyo, Japan). The specific parameters were: the scanning angle was ranged from 20°–80°, the scanning speed was 10°/min, and the target material was a copper target. The image analysis method and mercury intrusion method were used to analyze the material's pore structure's evolution. The density of the

samples was measured by the Archimedes method. The line shrinkage of the samples was calculated by the formula listed below:

$$\delta = \frac{L_0 - L_1}{L_0} \tag{1}$$

where $\delta$ is the line shrinkage of the sample, $L_0$ is the length of the designed model, $L_1$ is the length of sintered samples.

## 3. Results

### 3.1. Raw Powder Analysis

Figure 1 shows the size distribution and the scanning electron microscopy (SEM) of SiC and $B_4C$ powder applied in the SLS process. Figure 1a shows the powder granularity of the SiC powder, and the mean particle size of the SiC powder was 19.4 μm. While Figure 1b shows the morphology of $B_4C$, its mean particle size is also 19.4 μm. Both powders are irregular polygonal forms and show no aggregation.

In this study, a zigzag scanning strategy was used to sinter the powder bed of mixed powder (as shown in Figure 2). This strategy has little difference in stress distribution than single-direction line-by-line scanning, but this method is relatively simple. It can significantly reduce the temperature difference between the forming area and the powder bed [24]. Hence, it can alleviate the thermal stress and avoid ceramic cracking during direct laser selective melting and forming.

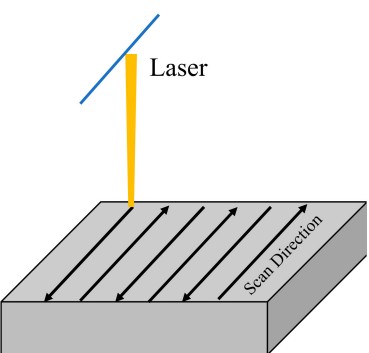

**Figure 2.** Scanning strategy applied in this study.

### 3.2. Forming Process Analysis

Figure 3 shows the processing window with no boron carbide addition and with 15 wt% $B_4C$ addition. It can be seen from the figure that the processing window with 15 wt% $B_4C$ addition is much broader. The processing window's expansion is related to the interaction between boron carbide and silicon carbide during laser sintering.

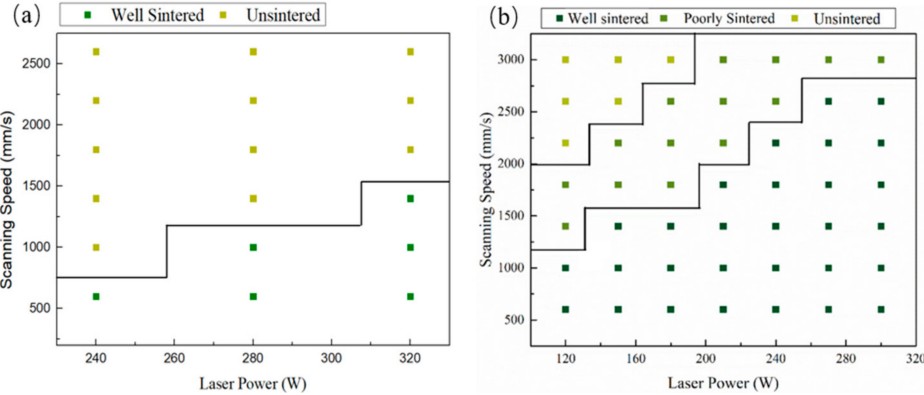

**Figure 3.** The processing window of SiC (**a**) with no $B_4C$ additionand (**b**)with 15 wt% $B_4C$ addition.

The boron carbide can form a high-temperature liquid phase with silicon carbide and generate a low melting point phase with Si formed by the decomposition of silicon carbide (as illustrated in Figure 4) [25]. The interaction among $B_4C$, SiC, and Si prolongs the amount and duration of the liquid phase. Therefore SiC can be laser sintered with lower laser energy density.

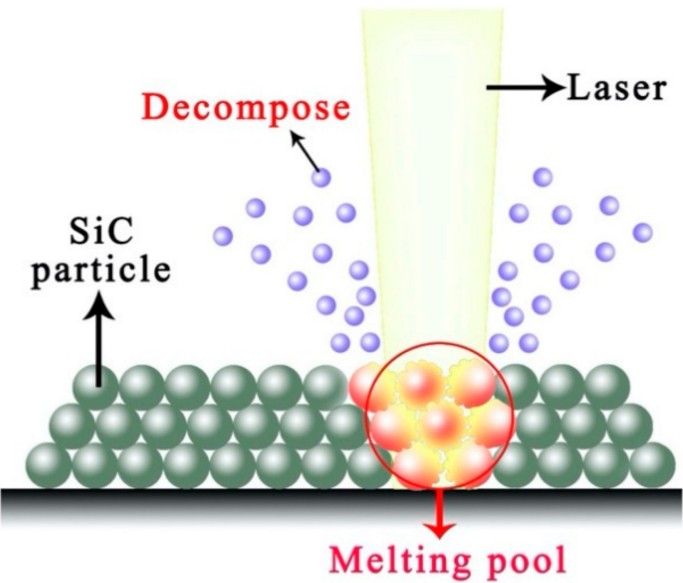

**Figure 4.** Laser–SiC interaction in the argon atmosphere.

This phenomenon is also manifested by the expansion of the range of energy density suitable for processing. The energy density can be calculated according to the formula shown below:

$$E_d = \frac{P}{vH} \qquad (2)$$

where: $E_d$ represents energy density ($J/mm^2$), $P$ represents laser power (W), $H$ represents hatch spacing (mm), and $v$ represents scanning speed (mm/s). Therefore, the minimum energy density for pure SiC is 457 $J/cm^2$. When the $B_4C$ addition reached 15 wt%, the minimum energy density for forming SiC was 214 $J/cm^2$. It is particularly worth mentioning that the available scanning speed with $B_4C$ addition was above 2000 mm/s, which meant a short interaction time between laser and mixed powder. As reported by Yin et al., the splash phenomenon is mainly related to the laser power, and the laser scanning speed determines whether the splash has enough time to occur [26]. It should be pointed out that the SiC shows an absorption coefficient close to 78% for the 1064 nm wavelength laser [27,28]. Therefore, the splash phenomenon of SiC ceramics will be a more severe splash phenomenon when the scanning speed is too slow.

The splashing phenomenon will be alleviated with a higher scanning speed. Therefore, this indicated that with the increase of boron carbide addition, due to higher scanning speed being adopted, the surface roughness of the sample will also be improved. This phenomenon is consistent with the phenomenon observed in the experiment. The samples prepared by a higher scanning speed ($P$ = 300 W, $V$ = 1800 mm/s) are shown in Figure 5. As can be seen from the figure, the square samples formed by direct laser sintering showed good integrity in terms of shape. A slight warp can be observed in the edge of all the samples; a tiny crack can be observed in larger samples shown in Figure 5a. That indicated the internal stress accumulated with the increase of the sample size. Therefore, the zigzag strategy is not suitable for the preparation of larger samples. The line shrinkage of the samples was about 4.8% when $B_4C$ addition reached 15 wt%. The shrinkage of the samples was only 1.2% when pure SiC was adopted. In our previous study on reaction-bonded $B_4C$/SiC, when the $B_4C$ addition reached over 15 wt%, the strength of the samples could dramatically

be decreased, and abnormal grain growth could be observed in the microstructure when the sintered temperature was over 1650 °C. This abnormal grain growth was also called coarsening during the silicon infiltration process into the $B_4C$ body (reported by Cuiping Zhang et al. [29]). Laser irradiation can heat the scanning area to more than 2000 °C; In our study, the highest $B_4C$ content was selected as 15 wt%.

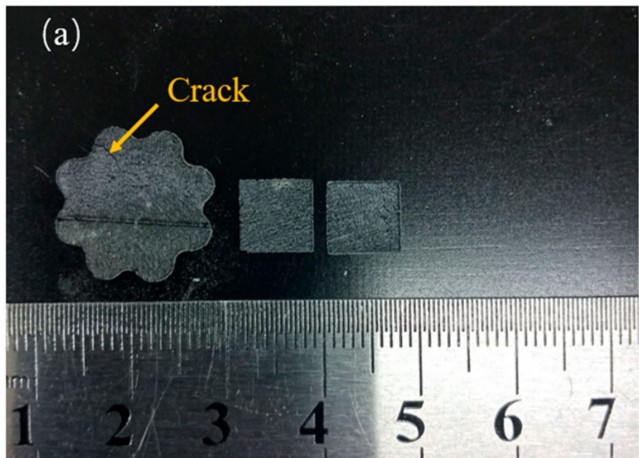
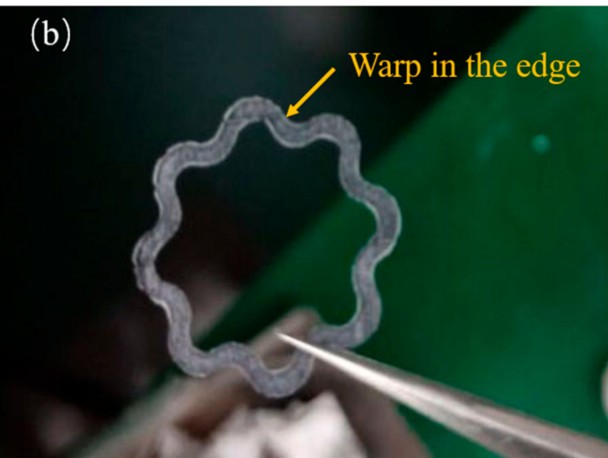

**Figure 5.** Porous SiC parts with 15% $B_4C$ addition fabricated by direct selective laser sintering (**a**) Square samples and larger samples with crack and (**b**) samples with a warped edge (*P* = 300 W, *V* = 1800 mm/s).

### 3.3. Phase and Microstructure Evolution

Figure 6 shows the phase composition of SLSed SiC with different $B_4C$ contents under 320 W, 800 mm/s. The amount of residual silicon is obtained by semi-quantitative analysis of X-ray diffraction (XRD) data. It can be seen from the figure that as the content of boron carbide increases, the characteristic peak of silicon gradually decreases. With no $B_4C$ addition, the amount of residual silicon is about 38 at.%. When the content of $B_4C$ reached 5 wt%, the amount of residual silicon decreased to about 10 at.%, which is about 8 vol% Si. In the study performed by Chakrabarti et al. [30], the mechanical performance of SiC with residual Si content of about 16.5% showed good high-temperature strength and toughness. It was about 30% higher than samples with 24% residual silicon. In the research performed by Meyers [21], they fabricated SiC with a lower silicon content of about 16 vol% and also showed good mechanical performance. Therefore, by the addition of $B_4C$, the properties of porous SiC could be significantly improved.

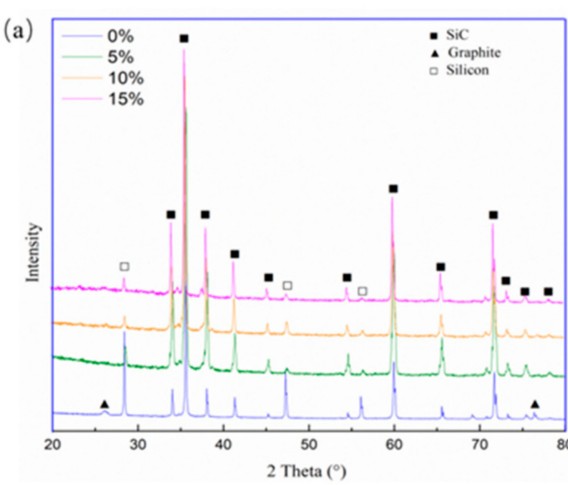
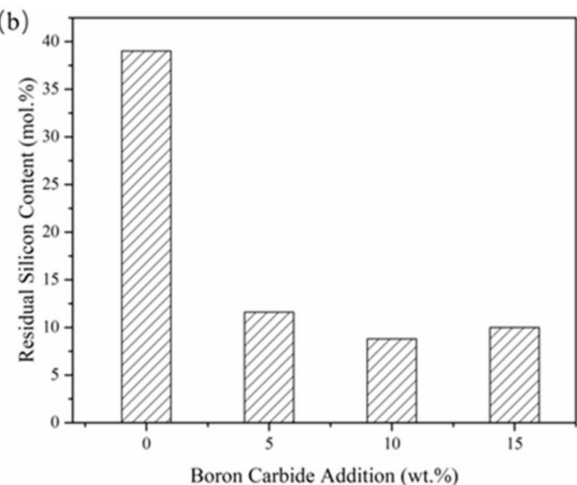

**Figure 6.** (**a**) X-ray diffraction (XRD) results with different $B_4C$ addition and (**b**) calculated residual silicon contents.

During the laser sintering process, the temperature of the laser-sintered area can reach over 3000 °C [31]. When the amount of $B_4C$ increased, the silicon content stabilized at about 8 at.%. The change of Si content is a little different from the traditional sintering process of $B_4C$ and Si. In the conventional sintering process, the reaction between Si and $B_4C$ is dominated by diffusion. However, in the laser sintering process, the reaction will be terminated as the laser sintering process is finished. Therefore, most of the $B_4C$ will react with Si and generate silicon borides (such as $SiB_4$, $SiB_6$, etc.) and SiC. However, due to the fast cooling rates during the laser sintering process, these silicon borides and other reactants are usually amorphous phases. Therefore it cannot be identified in the XRD pattern. The energy-dispersive X-ray spectroscopy (EDS) results showed its existence in the sintered body (Figure 7). The decomposition of the SiC process during laser sintering can be displayed as follow:

$$SiC \rightarrow C + Si \tag{3}$$

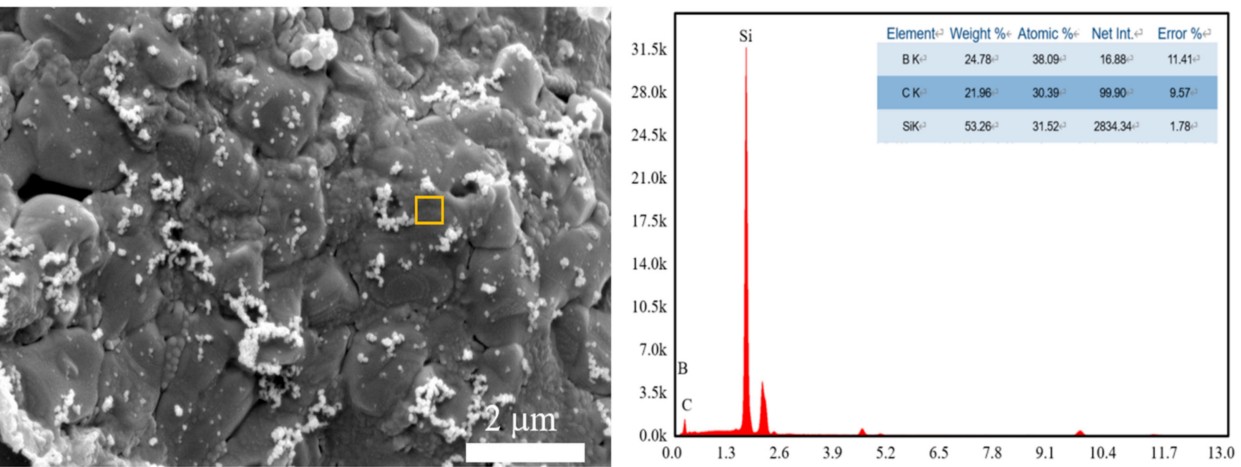

**Figure 7.** Energy diffraction spectra (energy-dispersive X-ray spectroscopy, EDS) results of porous SiC with 15 wt% $B_4C$ addition.

When pure SiC was laser sintered (P > 280 W), the decomposition of SiC occurred. The peaks at 26.38°, 76.38° are corresponding to C, and the peaks at 28.6°, 47.5°, 56.78° is corresponding to Si. It decreased with the addition of $B_4C$. This phenomenon should be attributed to the formation of the eutectic phase of boron carbide and silicon carbide, which suppressed the decomposition of SiC.

The reactants can also be deduced from the phase diagram of SiC-$B_4C$ and Si-$B_4C$(as shown in Figure 8 which can be found in http://www.msi-eureka.com/, accessed on 11 April 2021). As can be seen from Figure 8b, $B_4C$ can form a more liquid phase with Si when the temperature is higher than 1400 °C but could form a small amount of liquid phase with SiC under the same temperature. As the content of $B_4C$ increased, the Si content decreased. This difference is attributed to the reaction of $B_4C$–Si and the decomposition process of SiC. These two processes are both dominated by the laser sintering process. It indicated that even if the $B_4C$ content increases, there will always be a small silicon amount that $B_4C$ has not consumed in time.

The microstructure with different $B_4C$ addition is shown in Figure 9. As shown in Figure 9a,b, a porous structure with no clear orientation was formed by the stacking of SiC particles. A continuous silicon carbide struts like stripes can be observed in Figure 9c when $B_4C$ addition reached 10 wt%. When boron carbide addition reached 15 wt%, as shown in Figure 9d, an oriented porous structure can be observed, and large pores were formed. As shown in Figure 9d, the SiC and $B_4C$ particles were all melted together. This phenomenon is attributed to the generation of liquid phase content caused by the addition of $B_4C$ content. Because the generation of the liquid phase is strictly related to the laser sintering process, the capillary force generated by the liquid phase drives the particle to regroup during laser sintering and forms an oriented structure. Due to the addition of

B$_4$C enhancing the densification process of SiC [32], the porous struts with 15 wt% B$_4$C addition are denser than those with no B$_4$C addition, which would be beneficial for its mechanical performance.

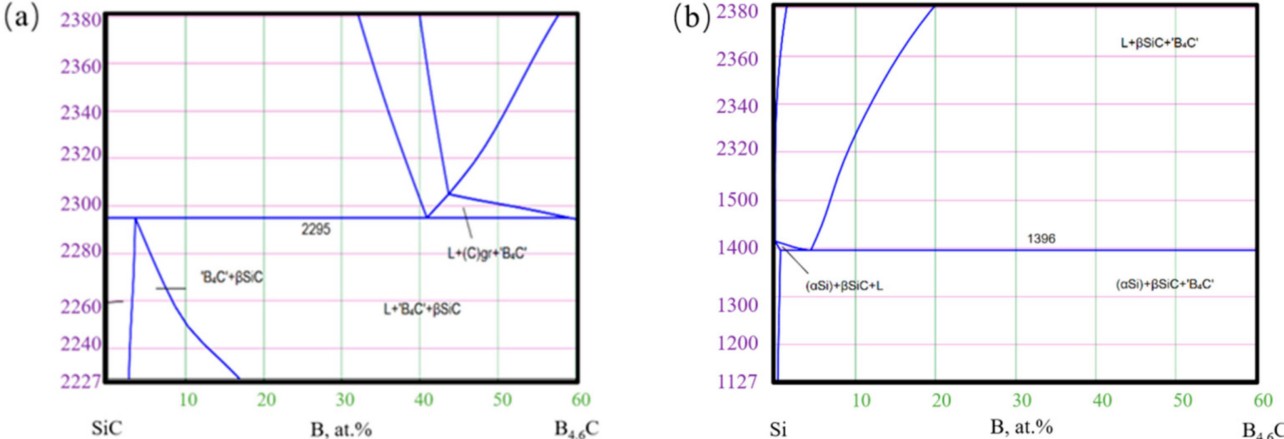

**Figure 8.** Phase diagram of SiC-B$_4$C (**a**) and Si-B$_4$C (**b**).

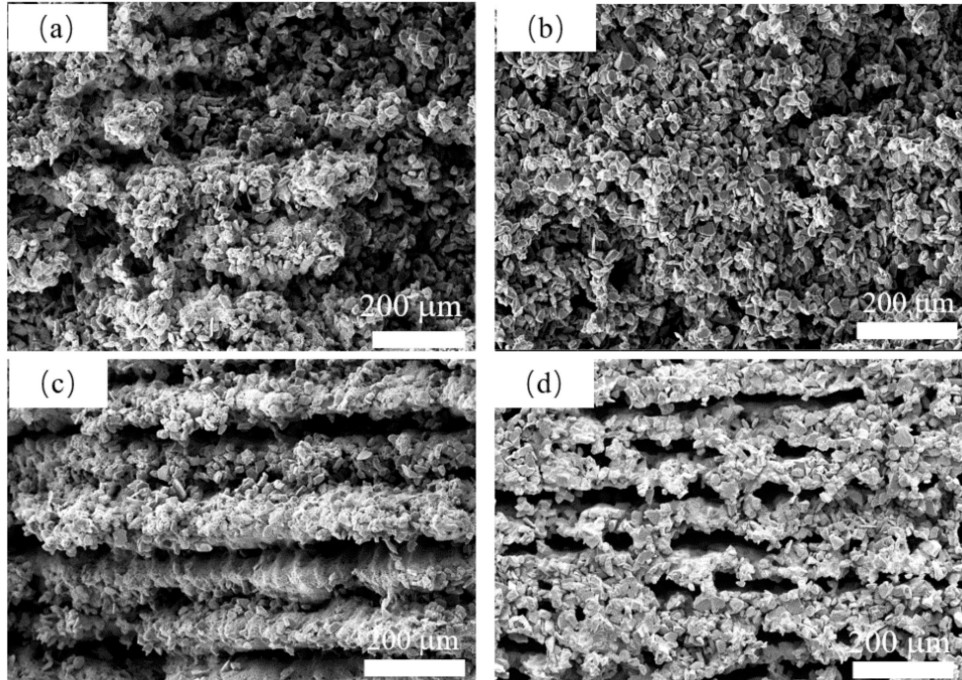

**Figure 9.** The SEM of the surface of the samples with (**a**) 0 wt% B$_4$C, (**b**) 5wt% B$_4$C, (**c**) 10 wt% B$_4$C and (**d**) 15 wt% B$_4$C under 320 W, 1000 mm/s.

### 3.4. Pore Size Analysis

The pore size distribution of porous SiC is shown in Figure 10. As shown in the figure, with the increase of B$_4$C addition, the number of bigger pores (about 9 μm) increased. The smaller pores (approximately 6 μm) decreased, indicating the increase of mean pore diameter. The mercury intrusion results showed that the mean pore diameter slightly increased from 9.5 μm to 11.3 μm. The results are consistent with the results of microstructure observation. With the addition of B$_4$C increased, the amount of liquid phase in the laser sintering process also increases, promoting densification process, which enhanced the tiny pores merge to grow into large pores. The porosity of the porous SiC with no B$_4$C addition was 53%, when the addition of B$_4$C reached 15 wt% the porosity can reach 67%, and the density is about 1.61 g/cm$^3$ and 2.03 g/cm$^3$, respectively.

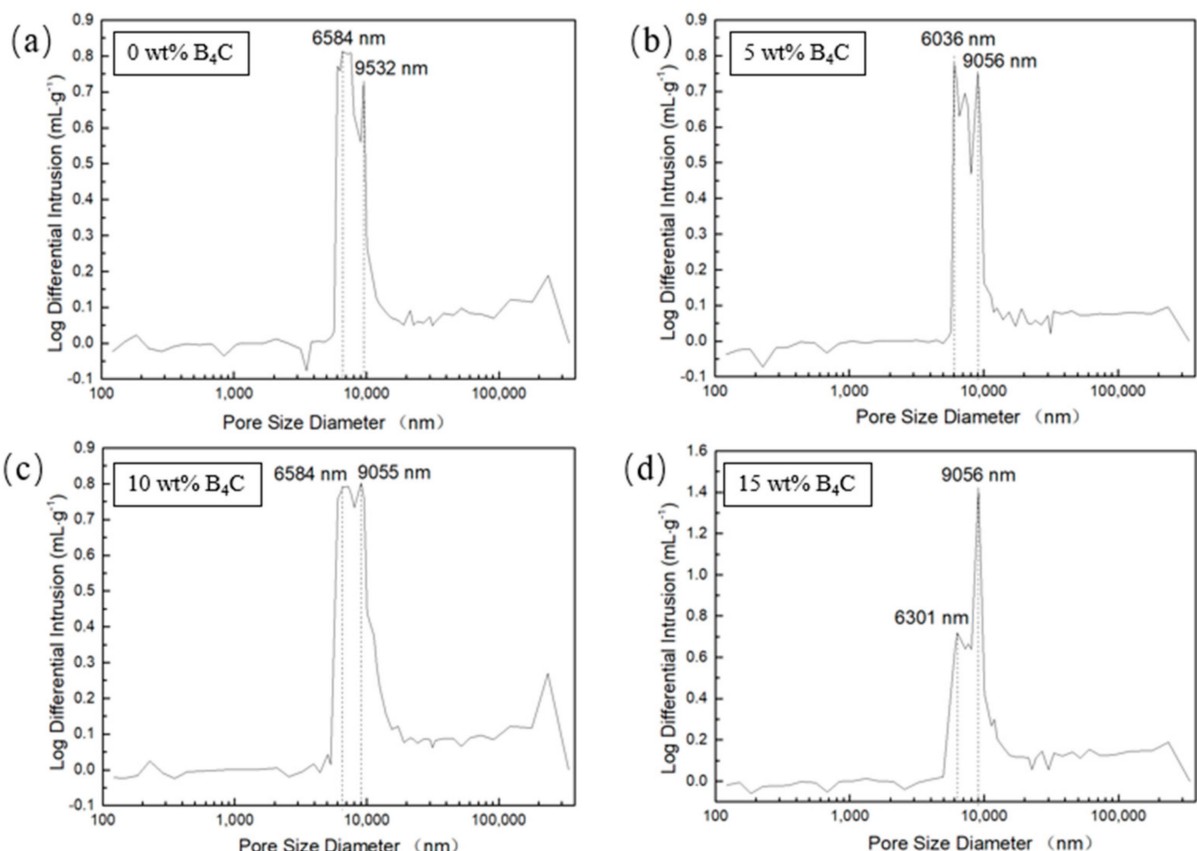

**Figure 10.** Pore size distribution of porous SiC fabricated under 320 W, 1000 mm/s,(**a**) with 0 wt% B$_4$C, (**b**) with 5 wt% SiC, (**c**) with 10 wt% SiC, (**d**) with 15 wt% SiC.

## 4. Conclusions

This paper evaluated the influence of the adoption of B$_4$C on porous SiC via the direct SLS method. The dSLS process could directly obtain porous SiC with low silicon content when B$_4$C was introduced. When the amount of B$_4$C added to the mixed powder increased, the process window was enlarged. The minimum energy density required for forming was reduced from 457 J/cm$^2$ to 215 J/cm$^2$.The microstructure observation showed that the increase in boron carbide content during the laser selective sintering process would make the particles in the sintered body melt more completely. The increase in the range of boron carbide will enhance the orientation of the structure. The mercury intrusion method results showed that with the increase of the boron carbide content, the porous material's tiny pores decreased, and the macropores increased, which is consistent with the morphological observation results. Future work should be undertaken to evaluate the influence of using finer powder particle sizes, the influence of B$_4$C content on improving the mechanical properties, and corrosion behavior.

**Author Contributions:** Conceptualization, R.L. and Y.S.; methodology, R.L.; validation, P.C., G.C. and R.L.; formal analysis, R.L., C.Y.; investigation, G.C., P.C. and Y.Q.; resources, Y.S. and H.T.; data curation, R.L. and P.C.; writing—original draft preparation, R.L.; writing—review and editing, R.L., C.Y. and G.C.; visualization, R.L. and Y.Q.; project administration, R.L., Y.S. and H.T.; funding acquisition, All authors have read and agreed to the published version of the manuscript.

**Funding:** This research was funded by the National Natural Science Foundation of China (Grant No. 51875222), China Postdoctoral Science Foundation (2017M622426), the First Class Special funding for Postdoctoral Scientific Research of Hubei Province (2017-G3), the Opening Fund of State key laboratory for Environment-friendly Energy Materials (17kffk 12).

**Data Availability Statement:** The data presented in this study are available on request from the corresponding author. The data are not publicly available due to all the dataset created during this research belong to the funder according to the contract.

**Acknowledgments:** The authors are also grateful for the State Key Laboratory of Materials Processing and Die and Mould Technology for mechanical property tests, as well as the Analysis and Testing Center of Huazhong University of Science and Technology for XRD and SEM tests.

**Conflicts of Interest:** The authors declare no conflict of interest. The funders had no role in the design of the study; in the collection, analyses, or interpretation of data; in the writing of the manuscript; or in the decision to publish the results.

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
