# Peer review of "Fabrication of Porous SiC by Direct Selective Laser Sintering Effect of Boron Carbide"

_metals, doi:10.3390/met11050737_

Round 1
Reviewer 1 Report
In presentet manuscript the authors analyzed a a influence of B4C on modification additive manufactured porous SiC. The itself SiC is a promising material applied in extreme conditions such as high temperature, chemical corrosion or irradiation. However, are a publications wich was stated that the residual Si's existence deteriorates its performance and limits its application in harsh environments. Therefore the authors B4C introduced into the selective laser sintering process of SiC, and its effects on forming ability, pore parameters, microstructure, and phases were investigated. Their research showed that when B4C was added, the processing window was enlarged. The addition of B4C increases the amount of liquid phase during the laser sintering process of silicon carbide, improving SiC struts' density and reducing the residual silicon by reacting with it. In presented manuscript the authors shows that the addition of B4C will help improve the application performance of laser sintered silicon carbide.
The study in manuscript raise an important aspect, and the research will expand knowledge in this field. However, there are some unanswered issue that need to explain by the authors.
- Please describe in more detail how the samples were prepared prior to the tests presented in "2.3 Characterization"
- The information on lines 125-127 should be deleted.
- In line 152 the authors wrote about a boron carbide can form a high-temperature liquid phase with silicon carbide and generate a low melting point phase with Si formed by the decomposition of silicon carbide. I propose provide melting points of these compounds.
- In Figure 5, the authors show two photos, but in the text of the manuscript, they did not comment on what is shown in Figure 5a and Figure 5b. The descripton Figure 5 is too generall.
- Figure 6a is illegible. Please present the drawings one below the other. In a vertical arrangement, they will be clearer and more readable.
- In the presented manuscript, the authors present only the results of microstructure where is addition of 15% B4C for forming selective laser sintering process of SiC. What changes in microstructure (please put photos) took place in the material after adding, for example, 5% or 10% B4C? Why are the most promising results at 15% B4C? Have the authors tried with more B4C for example 50%?
- How does the amount of B4C introduced affect the mechanical properties?Are there or have been such studies conducted by the authors?
Author Response
Thanks so much for your careful review work. All the revisions were marked in red in the manuscript. Here is a detailed response from point to point.
- Please describe in more detail how the samples were prepared prior to the tests presented in "2.3 Characterization"
In 2.2 and 2.3 new details of how the samples were fabricated and the size for further characterization have been supplied.
"
During laser sintering, the mixed powder was put into a bunker and then paved by the roller to form the powder layer. 316L Steel plate was adopted as a base plate. The powder was then laser sintered to form the porous sample. Square Samples with a dimension of 8 mm´ 8 mm with a thickness about 5 mm was fabricated to evaluate the integrity and line shrinkage during laser sintering process. After laser sintering, the samples were carefully removed from the base plate by wire cutting. The morphology and phase analysis was also performed by this type sample."
- The information on lines 125-127 should be deleted.
Thanks, It has been deleted right now.
- In line 152 the authors wrote about a boron carbide can form a high-temperature liquid phase with silicon carbide and generate a low melting point phase with Si formed by the decomposition of silicon carbide. I propose provide melting points of these compounds.
The phase diagram was provided by the authors right now.
- In Figure 5, the authors show two photos, but in the text of the manuscript, they did not comment on what is shown in Figure 5a and Figure 5b. The descripton Figure 5 is too generall.
- Figure 6a is illegible. Please present the drawings one below the other. In a vertical arrangement, they will be clearer and more readable.
The figure has been updated according to your suggestion.
- In the presented manuscript, the authors present only the results of microstructure where is the addition of 15% B4C for forming selective laser sintering process of SiC. What changes in microstructure (please put photos) took place in the material after adding, for example, 5% or 10% B4C? Why are the most promising results at 15% B4C? Have the authors tried with more B4C for example 50%?
The selection of 15% B4C is base upon our previous research on reaction bonded B4C/SiC composites. Too much B4C will lead to abnormal grain growth and deteriorate mechanical strength. The microstructure figure with different B4C addition is updated now and microstructure evolution is now discussed.
- How does the amount of B4C introduced affect the mechanical properties?Are there or have been such studies conducted by the authors?
We have not studied the mechanical performance of this porous material systematically yet. It will be our future work. The conclusion part has been modified accordingly.
Reviewer 2 Report
Please refer the attachment.

Author Response
We feel great thanks for your professional review work on our article. Here is the response from point to point.
Major points 1)p.3, l.105: Please show the materials of ball and container of ball-milling.
The materials of ball and ball containers have been provided in the manuscript.
2)p.3, l.105~: After ball milling, particle size and morphology of the raw powder is considered to be changed. These SEM images is required.
It's a great pity, due to this work has been done last summer, the authors did not keep the data after balling such as SEM or particle size distribution. The deadline of response is too short for us to provide the new data of these powders. We did characterization for the angle of repose test of the powder and kept the photo and data. If needed, we will provide it.
3)p.4, l.145: Please show the temperature information like thermographic data during dSLS to understand the sintering mechanism. If you do not have these data, you have to discuss the estimated temperature from phase diagram.
The phase diagram and estimated temperature information were provided in the manuscript right now
4)Figure 6(a): InXRD peaks of 0% B4C, the peaks around 56°and 26°were not assigned. What the materials is assigned to these peaks. In particular, the peak around 56°remained in the case of 5, 10% B4C. You should discuss this point.
We checked all the possible reactants and chemicals, finally we found the two peaks should be corresponding to different materials caused by the decomposition of SiC. The XRD results were updated.
"
During the laser sintering process, the temperature of laser-sintered area can reach over 3000 ℃[31] When the amount of B4C increased, the silicon content stabilized at about 8 at.%. The change of Si content is a little different from the traditional sintering process of B4C and Si. While in the traditional sintering process, the reaction between Si and B4C is dominated by the diffusion process. However, in the laser sintering process, the reaction will be terminated as the laser sintering process finished. Therefore, although most of the B4Cewillereact with Si and generate silicon borides (such as SiB4, SiB6, etc.) and SiC. However due to the fast cooling rates during the laser sintering process, these silicon borides and other reactants is usually amorphous phase, therefore it cannot be identified in XRD pattern. The EDS results showed its existence in the sintered body(as shown in Fig. 7). The decomposition of the SiC process during laser sintering can be shown as follow:
SiC=C+Si (3)
When pure SiC was laser sintered (P>280W), the decomposition of SiC occurred. The peak at 26.38°, 76.38° is corresponding to C)and 28.6°, 47.5°, 56.78° is corresponding to Si. It decreased with the addition of B4C. This should be attributed to the formation of the eutectic phase of boron carbide and silicon carbide, which suppressed the decomposition of SiC.
"
5)Figure 6(b): 8at% silicon residue seems like a lot, but is it a problem as a practical material? In this regard, the amount of residual silicon should be discussed in comparison with other research examples.
New references has been cited to compare with other research examples.
6)Has all the B4C been decomposed and all the boron sputtered out? If not, where is the remaining boron?
The B4C reacted with Si and form new liquid phases with SiC. Due to the fast cooling rates of laser sintering, these phases can not be determined by XRD results. We provided new EDS figure to illustrate it .
7)Density of the sintered body is required to evaluate this method is useful.
Now the density data of pure SiC and sample with 15wt% B4C addition is provided.
Minor points 1)p.1, l. 17:J/cm2 -> J/cm2
It has been rivised.
2)No Keywords showed.
The version is updated now. The keywords should be shown right now.
3)p.3, l.103: Figure 1 -> Figure 2?
The figure numbers were updated.
4)p.3, l.129-131: The same-meaning sentences was repeated.
This sentence has been deleted.
5)Figure 6(a): When printed, the figures are shaded and difficult to read.
The figure has been redrawn and modified according to other suggestions.

Round 2
Reviewer 1 Report
The authors responded to all my comments. I recommend publishing the manuscript.
Reviewer 2 Report
Sentence of l.276 is strange and please correct it.